# Bushfires and Mothers’ Mental Health in Pregnancy and Recent Post-Partum

**DOI:** 10.3390/ijerph21010007

**Published:** 2023-12-20

**Authors:** Nicolas Cherbuin, Amita Bansal, Jane E. Dahlstrom, Hazel Carlisle, Margaret Broom, Ralph Nanan, Stewart Sutherland, Sotiris Vardoulakis, Christine B. Phillips, Michael J. Peek, Bruce K. Christensen, Deborah Davis, Christopher J. Nolan

**Affiliations:** 1National Centre for Epidemiology and Population Health, Australian National University, Canberra, ACT 2601, Australia; 2School of Medicine and Psychology, Australian National University, Canberra, ACT 2601, Australia; amita.bansal@anu.edu.au (A.B.); christopher.nolan@anu.edu.au (C.J.N.); 3John Curtin School of Medical Research, Australian National University, Canberra, ACT 2601, Australia; 4The Canberra Hospital, Canberra Health Services, Garran, ACT 2605, Australia; hazel.carlisle@act.gov.au (H.C.);; 5School of Nursing and Midwifery, University of Canberra, Bruce, ACT 2617, Australia; deborah.davis@canberra.edu.au; 6Sydney Medical School and Charles Perkins Centre Nepean, University of Sydney, Penrith, NSW 2750, Australia; ralph.nanan@sydney.edu.au; 7Nursing and Midwifery Office, ACT Government Health Directorate, Phillip, ACT 2606, Australia

**Keywords:** pregnancy, bushfires, COVID, mental health, wellbeing

## Abstract

Background: The compounding effects of climate change catastrophes such as bushfires and pandemics impose significant burden on individuals, societies, and their economies. The enduring effects of such syndemics on mental health remain poorly understood, particularly for at-risk populations (e.g., pregnant women and newborns). The aim of this study was to investigate the impact of direct and indirect exposure to the 2019/20 Australian Capital Territory and South-Eastern New South Wales bushfires followed by COVID-19 on the mental health and wellbeing of pregnant women and mothers with newborn babies. Methods: All women who were pregnant, had given birth, or were within three months of conceiving during the 2019/2020 bushfires, lived within the catchment area, and provided consent were invited to participate. Those who consented were asked to complete three online surveys. Mental health was assessed with the DASS-21 and the WHO-5. Bushfire, smoke, and COVID-19 exposures were assessed by self-report. Cross-sectional associations between exposures and mental health measures were tested with hierarchical regression models. Results: Of the women who participated, and had minimum data (*n* = 919), most (>75%) reported at least one acute bushfire exposure and 63% reported severe smoke exposure. Compared to Australian norms, participants had higher depression (+12%), anxiety (+35%), and stress (+43%) scores. Women with greater exposure to bushfires/smoke but not COVID-19 had poorer scores on all mental health measures. Conclusions: These findings provide novel evidence that the mental health of pregnant women and mothers of newborn babies is vulnerable to major climate catastrophes such as bushfires.

## 1. Introduction

The frequency of extreme environmental events such as bushfires and pandemics is on the rise globally, and is predicted to worsen in intensity, frequency, and length with our changing climate [1]. It is evident that the increasing frequency and intensity of bushfires and concurrent or subsequent infectious disease outbreaks confer a global societal, public health, and economic burden. Our knowledge of the long-term effects of such syndemics on mental health is still evolving [2]. Understanding this is critical to fostering preparedness for future events and to mitigating their debilitating impact on the health of affected individuals and the health system.

Bushfires expose larger populations to extreme heat and smoke-related air pollution, and have been reported to adversely affect cardiometabolic and respiratory health [3,4,5,6,7] and increase mortality [8,9]. Existing evidence confirms significant impacts on mental health, including post-traumatic stress symptoms, anxiety, depression, and impaired wellbeing [10]. However, the short- and long-term effects of bushfires on the mental health of pregnant women and mothers with babies during and after a bushfire crisis remain poorly understood and under-investigated [11]. This is concerning because we already know that this population is particularly vulnerable to impaired mental health in the absence of additional environmental stressors, with systematic reviews estimating the prevalence of perinatal depression at 12% for major depressive disorder [12] to 25–27% for more inclusive depression definitions [13]. 

There is evidence, mainly using cross-sectional studies and linked data, that bushfire exposure adversely affects pregnancy and birth outcomes and increases morbidity for childbearing women and their babies [14,15,16,17,18,19,20,21]. In addition, it is widely recognised that natural disasters increase prenatal maternal stress and impair neurodevelopment in children [22,23,24,25]. A recent systematic review found that those exposed to natural disasters, including tsunamis, floods, hurricanes, earthquakes, wildfires, landslides, volcanic eruption, and typhoons were on average almost twice as likely to experience psychiatric disorders and showed significantly higher psychological distress [26]. The impacts of natural disasters on mental health appear not to be transient and can last for years, with children being particularly affected [27]. However, it should be noted that there is high heterogeneity between studies in this field, with estimates of the prevalence of depression following natural disasters ranging from 6–54% in adults and 8–45% in children [28]. Unfortunately, we currently lack evidence regarding the impact of natural disasters such as extreme bushfire events experienced during preconception, pregnancy, and postpartum on maternal mental health or childhood neurodevelopment, which limits our capacity to develop evidence-based strategies to mitigate these risks and provide the best support when they occur. 

The 2019/2020 Australian bushfires were unprecedented in duration, severity, and scale [29,30], and the Australian Capital Territory (ACT) and South Eastern-New South Wales (SE-NSW) regions were among the worst affected. The severity of maternal exposure to bushfire smoke was much higher (the average ambient concentration of fine particulate matter, PM_2.5_, was ~124 μg/m^3^ in the Tuggeranong valley of the Australian Capital Territory, over 57 days) [31] than reported in previous wildfire studies, where it has ranged from ~4.4 to 57 μg/m^3^ [16,17]. The hourly levels of PM_2.5_ were >1000 μg/m^3^ on the worst days. A recent cross-sectional study of 2084 adults (median age 45 years old) from this region reported negative effects of exposure to the 2019/2020 Australian bushfires on the physical and mental health and sleep patterns of the participants [32]. Importantly, these bushfires were subsequently followed by the COVID-19 pandemic, such that back-to-back crises left little time for recovery. This is particularly concerning when we consider the compounding effects of bushfires and COVID-19 on health [33,34]. The restrictions imposed during the COVID-19 pandemic disproportionately impacted women and babies worldwide, including disruption of maternity and child healthcare services, social cohesion, family support, and women and their partners juggling work from home, house chores, and childcare at home [35,36,37].

To better understand how bushfires impacted the health of mothers and development of their babies and to assess how the COVID-19 pandemic may have compounded these effects, we established a longitudinal population-based cohort study called the Mother and Child 2020 (MC2020) study. It aimed to survey pregnant women and mothers with newborn babies who were exposed to bushfires and the COVID-19 pandemic and to follow the progress of mothers and their babies. In the current cross-sectional study, we sought to specifically investigate the associations between exposure to these synergistic crises (measured retrospectively) and the mental health of the mothers of the MC2020 cohort who were either pregnant, had given birth, or were planning a pregnancy at the time of the 2019/2020 bushfires in the ACT and SE-NSW.

## 2. Materials and Methods

This study adheres to the Strengthening the Reporting of Observational Studies in Epidemiology (STROBE) guidelines [38].

### 2.1. Study Population

Participants were drawn from the Mother and Child 2020 (MC2020) study from the ACT and SE-NSW region for whom births occurred between 1st November 2019 and 31st December 2020. To maximise participation and representativeness, the inclusion criteria only required the mother to be domiciled in the catchment area and within three months of conceiving, pregnant (all trimesters), or being within three months of childbirth on 1st February 2020. Women were invited to participate via an intensive communication strategy using mainstream media, social media, and health service providers. Those who expressed interest in participating were given access to an online form providing information on the aims and scope of the study and were invited to fill in a consent form if they met the inclusion criteria. When consent was received, the participants were sent a link to the first “Bushfire” survey, which they could fill in at their own pace. Two weeks later they were invited to complete the follow-up “COVID-19” survey, and another two weeks later if their baby was more than three months old, or when the baby reached three months of age, they were asked to complete the “Pregnancy” survey. No compensation of any kind was offered for participation in the study.

Between 25th August 2020 and 31st December 2020, 1088 mother–child pairs consented to participate in the study and contributed sufficient data to be included. Of these, 985 completed the “Bushfire” survey, 869 completed the “COVID-19” survey, 738 completed the “Pregnancy” survey, and 635 completed all three surveys. After excluding participants who did not complete the bushfire measures (*n* = 28) or at least one of the mental health and wellbeing measures (*n* = 38), 919 participants were included in the analysis.

### 2.2. Bushfire and Smoke Exposures

Bushfire exposure was assessed through fourteen questions (Table 1) investigating whether participants had experienced certain events/circumstances (yes/no). An acute bushfire exposure measure was computed as the equally-weighted sum of self-reported possible or actual exposure-related events using eight questions likely to induce greater distress, including: being evacuated, being involved in fighting bushfires, having home or possessions destroyed, having personally suffered injuries due to bushfires, having a friend or relative injured due to bushfires, owning an animal that suffered as a result of the bushfires, and feeling frightened or upset during the bushfires. A broader exposure measure was computed by summing responses to the above items with those of another six questions relating to exposures less likely to induce distress.

Smoke exposure was assessed through a single self-reported question asking participants to rate the extent to which they had been exposed to smoke during the 2019–2020 bushfires (Table 1). Response options were “Not exposed”, “Minimally exposed—Less than 7 days with smoke just noticeable”, “Mildly exposed—More than 7 days with smoke just noticeable and less than 7 days with smoke easily noticeable”, “Moderately exposed—More than 7 days with smoke easily noticeable but less than 7 days of heavy bushfire smoke exposure”, “Severely exposed—More than 7 days of heavy bushfire smoke exposure” (not, minimally, mildly, moderately, severely). Responses from those reporting no, minimal, or only mild smoke exposure were combined to achieve a category of sufficient size (6.7%).

### 2.3. COVID-19 Exposure

COVID-19 exposure was assessed using the same approach as in our published study on a representative sample of the Australian population [39] based on ten questions (Table 1). An acute COVID-19 exposure measure was computed as the sum of self-reports of four potential or actual exposures to the virus likely to induce greater distress, including: having been diagnosed with the virus, having a family member diagnosed with the virus, having been directed to isolate, or having a family member who was directed to isolate. A broader COVID-19 exposure measure was computed by summing responses to the above items with those of another six questions relating to exposures less likely to induce distress.

### 2.4. Pregnancy

Perinatal phase (not pregnant, pregnant, new baby) during the bushfires was assessed by self-report at the time of the first survey. Pregnancy stage (trimester) during the bushfires was determined based on the date of the first survey and post-partum report of the delivery pregnancy week collected during the “Pregnancy” survey.

### 2.5. Mental Health and Wellbeing Measures

Symptoms of depression and anxiety over the last two weeks were assessed in the “Bushfire” survey with the 21 item version of the Depression Anxiety and Stress Scales (DASS-21) [40]. It is composed of three self-report sub-scales aimed at measuring negative emotional states of depression, anxiety and stress. The depression sub-scale is composed of seven items that measure symptoms specific to unipolar depression, the anxiety sub-scale is composed of seven items assessing generalised anxiety disorder symptomatology, and the stress sub-scale is composed of seven items assessing chronic non-specific arousal. Respondents use a 0–3 scale to indicate the extent to which each symptom applied to them in the past week, then sub-scale and total scores are computed by summing the scores of the relevant items and multiplying them by two to ensure consistency with the original 42-item version. The DASS-21 measures align closely with diagnostic criteria for major depressive disorder and generalized anxiety disorder [41,42,43], and the psychometric qualities of the scales have been extensively demonstrated [44], including post-partum [45]. 

General psychological wellbeing over the last two weeks was measured using the World Health Organization Wellbeing Index (WHO-5) [46]. The WHO-5 is a short self-reported measure of current mental wellbeing based on five items probing how the respondent felt in the last week (e.g., “I have felt cheerful in good spirits”) that is rated on a 0–5 point scale (“0: at no time” to “5: All the time”) and scored by summing all items and multiplying by four to obtain a final score ranging from 0–100. 

### 2.6. Covariates

Socio-demographic measures, including age, education level, parity (number of times participant has given birth), household income (categorised into low vs. middle/high), ever smoker (yes/no), hypertension (yes/no), and history of depression and anxiety were assessed by self-report. General health was assessed by self-rating using the first question of the SF36 (“In general, would you say your health is…: poor/fair/good/very good/excellent?”) [47].

### 2.7. Statistical Analysis

Statistical analyses were computed using the R statistical package (version 4.1.2, R Core Team, Vienna, Austria) under RStudio (version 2022.07.2 + 576, RStudio Team, Boston, MA, USA). Group differences were tested using Chi-square tests for categorical data and t-tests for continuous variables. Hierarchical regression analyses were conducted to test the associations between bushfire and smoke exposure and mental health and wellbeing measures. A first model only controlled for key covariates, including age, education, parity, and phase in pregnancy cycle. A second model also controlled for COVID-19 exposure. Additional models controlling for a greater number of covariates (household income, ever smoker, hypertension, history of anxiety and depression) were implemented as well. Logistic regression analyses were conducted to quantify the increased risk of poorer mental health outcomes associated with defined exposures (e.g., high bushfire exposure and past depression vs. not). Sensitivity analyses were conducted to determine whether differences in associations between bushfire exposure and mental health could be detected in relation to pregnancy stage (trimester), history of depression and anxiety, or co-habitation with partner/spouse by testing binary interactions between bushfire exposure pregnancy state, history of depression/anxiety, or co-habitation with partner/spouse. Bonferroni corrections were applied to adjust for multiple comparisons and the alpha level was set at <0.0125 (0.05/4).

## 3. Results

Participants’ characteristics are presented in Table 2. Participants were generally highly educated, predominantly from urban settings (90.2%), with good to excellent health (83%), and mostly pregnant or had delivered for the first time (86.4%). Of the 919 mothers included, 7.2% were not yet pregnant, 69.2% were pregnant, and 23.6% had a newborn baby at the time of the bushfires. Very few socio-demographic differences were detected between pregnancy status groups, although mothers who had already given birth were just over one year older than those who were pregnant or planning to have a baby.

Because our recruitment strategy is likely to have involved a selection bias, as participants were not randomly selected from the population, we used data from the Australian Bureau of Statistics (ABS) 2022 “Education and Work” and 2021 “Employee Earnings and Hours” surveys [48,49] including women between 25–34 years living in the Australian Capital Territory. Compared to ABS estimates, the participants had a lower proportion of tertiary education (64.4% vs. 69.5%), greater middle/high income (96.4% vs. 73.4%), and higher parity (0.73 vs. 0.57). These differences indicate a moderate amount of bias, which makes the present sample somewhat but not fully comparable to the population from which it was drawn.

### 3.1. Mental Health

Approximately 17.4% of participants reported having received treatment for depression and 34.2% treatment for anxiety prior to their pregnancy. No significant differences in WHO-5 and DASS-21 depression, anxiety, and stress sub-scale scores were detected between pregnancy phases (Table 2). MC2020 DASS-21 scores are contrasted to those of Australian pregnant women (*n* = 359; mean age 27.8 years, SD = 6.4) who experienced two or more adversity risk factors (e.g., living alone, poor health, low education, financial stress, etc.) in 2013 [50] and to Australian adult population norms (*n* = 497; 44% female; mean age 42.1 years, SD = 17.9, range 18–86 years) [51] in Table 3. MC2020 women had similar depression scores, lower anxiety scores (−32%), and higher stress scores (+6%) than women exposed to adversity risk factors. In contrast, MC2020 women had substantially higher depression (+12%), anxiety (+35%), and stress (+43%) scores relative to the general Australian adult population. Bivariate Pearson correlations between mental health measures, and bushfire and COVID-19 exposures are presented in Figure 1.

### 3.2. Bushfire and Smoke Exposure

The proportions of women reporting different types of bushfire exposures are reported in Figure 2. The exposures most frequently reported were being “frightened”, “relatives’ homes damaged”, “health being impacted” by bushfires, and “being evacuated” due to bushfires. More than 75% of participants reported one or more acute bushfire exposures, and 18% reported four or more exposures of any type. In addition, 30% reported moderate, and 63% severe smoke exposure.

### 3.3. COVID-19 Exposure

The proportions of women reporting different types of COVID-19 exposures are reported in Figure 2. The exposures most frequently reported were “testing negative to COVID-19 (due to potential exposure to virus)”, “knowing someone who had to isolate”, “voluntary isolation”, and “family isolating” due to COVID-19. More than 33% of participants reported one or more acute COVID-19 exposures, and 19% reported four or more COVID-19 exposures of any type.

### 3.4. Bushfire and Smoke Exposure and Mental Health

The main associations between bushfire and smoke exposures and mental health and wellbeing measures are presented in Table 4. Both higher acute and broad bushfire exposures were strongly associated with higher depression, anxiety, and stress symptomatology as well as with lower wellbeing ratings. Further analyses indicated that compared to women who had a low acute bushfire exposure (0–1), those who had a high exposure (>4) were much more likely to be classified as having moderate to extreme depression (OR: 4.42; 95%CI: 1.36–12.42; *p* = 0.007) or anxiety (OR: 5.11; 95%CI: 1.62–15.03; *p* = 0.004). In contrast, higher smoke exposure was not associated with higher scores on mental health measures; however, a trend was observed for lower wellbeing. Associations remained essentially the same when fully controlled models were conducted (Appendix A). 

### 3.5. COVID-19 and Mental Health

To determine whether COVID-19 direct and indirect exposure contributed to or moderated the associations detected between bushfire/smoke exposures and mental health and wellbeing measures, the above analyses were repeated on a somewhat reduced sample of participants (*n* = 777) for whom COVID-19 measures were available. Regression analyses controlling for acute and broad COVID-19 exposure are presented in Appendix A. Associations between broad COVID-19 exposure and mental health/wellbeing outcomes were moderately significant for DASS-21 depression, anxiety, and stress but not for the WHO-5 nor when only acute COVID-19 exposures were considered. In addition, while the associations between bushfire exposures and mental health measures were somewhat weaker after controlling for COVID-19 exposures, they followed the same pattern and remained highly significant. Similarly, associations between smoke exposure and mental health outcomes remained weak and mostly non-significant after controlling for COVID-19 exposure. To determine whether COVID-19 exposure moderated the effect of bushfire exposure, additional models including interaction terms were tested (Appendix A); however, these revealed no significant interactions.

### 3.6. Sensitivity Analyses

The possibility that other characteristics may have moderated the effect of bushfire exposure was tested for stage of pregnancy, history of depression and anxiety, parity, cohabitation with partner/spouse, household income, and setting (urban vs. rural).

Stage of pregnancy: To determine whether associations between bushfire exposure and mental health/wellbeing outcomes varied in pregnant women at different stages of pregnancy (*n* = 777), the same models tested in the whole sample were repeated while excluding those women who were not yet pregnant or had already given birth (Appendix A). No significant interaction effects between bushfire exposure and stage of pregnancy were detected, indicating that the associations between bushfire exposure and mental health did not vary at different stages of pregnancy.

History of depression and anxiety: Additional models testing associations between bushfire exposure and mental health/wellbeing outcomes were implemented to determine whether a history of prior depression or prior anxiety modulated these effects (Appendix A). The results showed that while past history of depression or anxiety was strongly associated with poorer mental health outcomes across all measures by 16–48%, it did not predispose to increased bushfire vulnerability.

Parity, cohabitation with partner/spouse, household income, setting: Additional models were implemented to test whether parity, living with a partner/spouse, household income (low vs. middle/high), or setting (urban vs. rural) moderated the associations between bushfire exposure and mental health outcomes (Appendix A). While history of depression and anxiety, higher parity, and lower education were associated with poorer mental health outcomes, no evidence that any of the measures investigated moderated the associations between bushfire exposure and mental health was detected.

## 4. Discussion

Three main findings emerged from this study. First, a large proportion of women who were planning a pregnancy, were pregnant, or had recently delivered at the time of the 2019/2020 bushfires reported high levels of bushfire and/or smoke exposure. These exposures were significantly associated with poorer mental health and wellbeing. Second, MC2020 participants had higher depression, anxiety, and stress levels compared to the general adult Australian population. Third, COVID-19 exposure was minimally associated with participants’ mental health, and did not substantially moderate the effect of bushfire exposure.

It is striking that three quarters of participants reported having been exposed directly or indirectly to the bushfires (e.g., family and friends being affected) and two thirds to severe smoke levels, while almost one in five women reported having experienced four or more bushfire exposure events. This may in part be due to a biased sample selection (i.e., those with greater exposure are more likely to volunteer for the study); nevertheless, it is likely to reflect the intensity with which bushfires were experienced in the regions we investigated. Importantly, greater exposure was associated with greater depression and anxiety symptomatology and lower wellbeing. Indeed, compared to women who had a low acute bushfire exposure (0–1), those who had a high exposure (>4) had a more than four-fold risk of being classified as having moderate to severe depression and anxiety. However, a similar effect was not observed for those who experienced smoke exposure. This is somewhat surprising, as it might have been expected that pregnant women would be particularly concerned about the effect smoke inhalation could have on the health and development of their baby. It is possible that the effects and media coverage of the most acute bushfire exposures involving more direct threat to individuals, their families, and their possessions might have distracted from longer-term concerns relating to smoke exposure. Alternatively, as a large proportion of women were exposed to smoke, albeit at different levels and for different durations, it is possible that even low or brief smoke exposure led to relatively similar concerns. If this were the case, then the lack of significant effects for smoke exposure might reflect widespread rather than non-existent concern. 

It is worth noting that with the exception of smoke, relatively few women (<5%) came into close proximity with bushfires. The most widely reported types of bushfire exposures were related to being frightened, damage to relatives’ home, impacts on health and healthcare, and being evacuated. This suggests that psychological effects and health concerns, rather than direct exposure and injury, might have particularly contributed to the participants reporting having experienced distress and anxiety. Such effects have been observed and discussed in the context of other types of traumatic events and natural disasters [52]; however, further research is needed in order to better understand the relative contribution of direct compared to indirect exposures.

Importantly, the mental health profile observed in MC2020 participants (bushfire exposed) was somewhat similar to that previously observed in pregnant Australian women exposed to adversity risk factors [50] (other than bushfires), albeit with somewhat lower anxiety, being substantially higher than Australian norms for this age group [51]. We acknowledge that some caution in interpreting these differences should be applied due to exposure and/or socio-demographic differences between these cohorts. For example, the Australian norms were developed between 1995–2000 and may be less applicable, particularly as the COVID-19 pandemic is likely to have impacted the mental health of the population not exposed to bushfires. Indeed, Takubo and colleagues [53] compared the mental health of women at one month post-partum assessed before or during the COVID-19 pandemic. They found that anxiety levels were higher in women assessed during the pandemic, while depression levels were lower than in those assessed prior the pandemic. Nonetheless, these findings are concerning, as pregnancy is already associated with a substantial risk of post-partum depression, with more than 11% of Australian women being affected after the birth of their child [54]; exposure to bushfires is likely to further increase this burden [11]. Moreover, mothers’ mental health, and specifically post-partum depression, are known to be associated with a child’s development, emotional wellbeing, and behavioural problems in both early childhood and in adolescence [55,56]. Follow-up of the MC2020 cohort is planned in order to explore the impact the bushfires and maternal mental health have on child health outcomes, including development. 

A somewhat surprising finding was that while women with a past history of depression and/or anxiety presented with significantly worse mental health, as would be expected based on the literature [57], this did not predispose them to reporting being more adversely affected by bushfire exposure. In addition, there was no evidence suggesting that women who did not live with a partner/spouse or who had low household income were more likely to report being affected by bushfires. Similarly, no associations with setting (urban/rural), parity, or trimester were detected. 

Another environmental influence that may have aggravated or better explained the lower mental health observed in the MC2020 cohort of women is the COVID-19 pandemic [39,53,58]. Analyses that considered bushfire and COVID-19 exposures together revealed non-significant associations between higher COVID-19 exposure and poorer mental health outcomes. This is somewhat inconsistent with findings from several previous studies that have reported a negative impact of COVID-19 exposure on mental health in the general population [58] as well as in pregnant women [59] even during the very early stages of the pandemic [60]. Notably, contradictory results in an Australian cohort have been reported as well [39]. However, in the present study the associations between COVID-19 exposure and mental health were independent of the bushfire effects, which suggests that they were additive and are associated with worse outcomes for women who experienced them together.

The fact that bushfire and COVID-19 exposures were assessed retrospectively should be acknowledged. However, due to the nature of the exposures and the type of measures used (e.g., “Were your own home, possessions or workplace damaged or destroyed?”; see Table 1), it is not likely they would have been substantially affected by recall bias in this population. Indeed, similar methods have been used to investigate associations between these exposures and mental health in other populations [39,61].

### Limitations

While significant effort was directed at systematically recruiting women in all stages of pregnancy exposed to bushfires within the catchment area, those in urban settings, with higher levels of education, were English speaking, and had higher socio-economic circumstances were over-represented. Moreover, while the study methodology strived to be inclusive of first nations women, different cultures, ethnicities, and minority groups, participant numbers in these groups were too small to contrast them to the rest of the cohort. The recruitment strategy is likely to have introduced other selection biases, although comparisons with sociodemographic figures from the Australian Bureau of Statistics provides some confidence that our sample is broadly representative. Nonetheless, this is the largest cohort of its kind internationally. The relatively large sample size ensured that sufficient statistical power was available to detect moderate effect sizes. However, it may not have been sufficient to detect interactions. Moreover, due to the correlational nature of the research, no causal associations can be inferred, and the poorer mental health outcomes observed in this population might be related to other factors. However, we took particular care in conducting additional analyses controlling for a large number of socio-demographic and health variables. Of note, only one of the MC2020 women was diagnosed with COVID-19 during the study period. Moreover, the peak of the pandemic was not experienced in the ACT until several months later, and the relative burden of lockdowns was experienced less severely overall in the investigated region than in some other areas of Australia.

## 5. Conclusions

In conclusion, the present findings indicate that the mental health of pregnant women and mothers of newborn babies is vulnerable to bushfire and smoke exposure. Consequently, the development of interventions to increase awareness of this risk before such disasters occur and the implementation of support structures to mitigate this risk during and after bushfires should be considered, particularly in the context of intensifying bushfires due to climate change.

## Figures and Tables

**Figure 1 ijerph-21-00007-f001:**
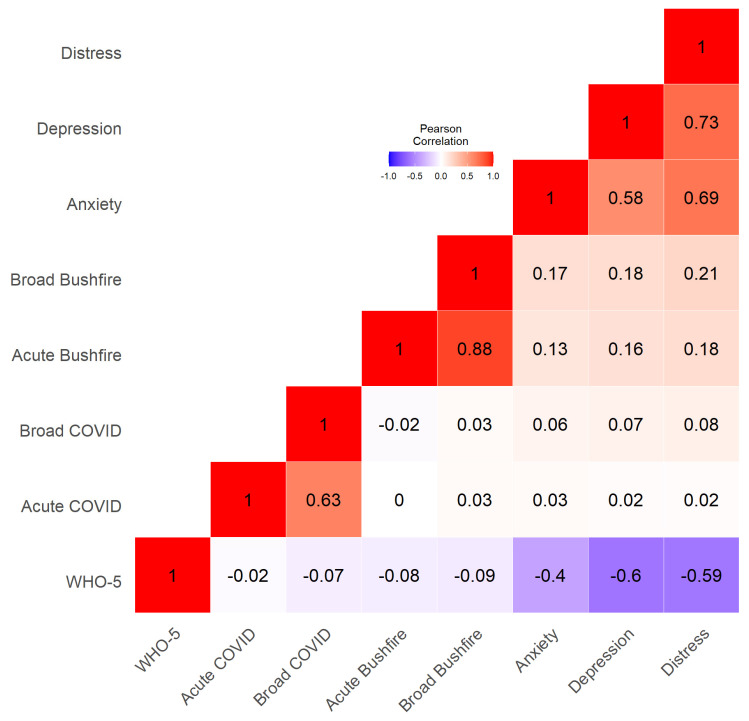
Bivariate associations between the main study variables. Bushfire and COVID-19 exposure measures were assessed by self-report. Acute exposure measures were computed based on items more likely to induce greater distress, while broad exposure measures were based on all available items. Smoke exposure (not, minimally, mildly, moderately, severely) was assessed through a single question. Mental health outcomes were assessed with the DASS-21 (depression, anxiety, distress) and the WHO-5 (wellbeing).

**Figure 2 ijerph-21-00007-f002:**
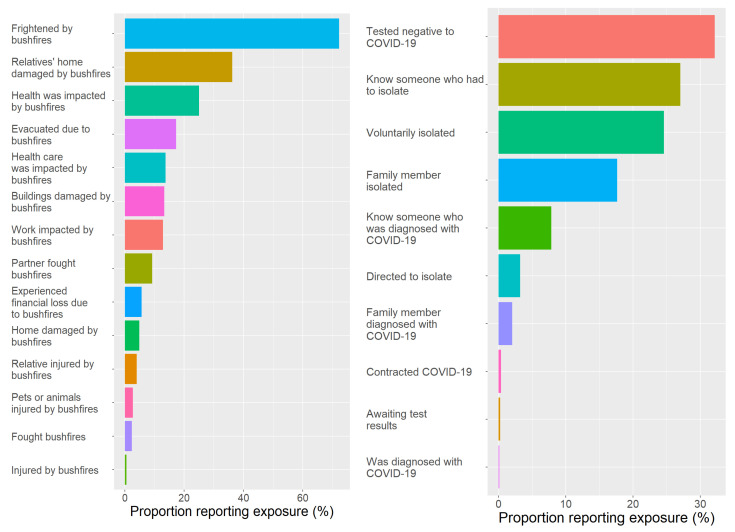
Proportions of women reporting different types of bushfire (**left**) and COVID-19 exposures (**right**).

**Table 1 ijerph-21-00007-t001:** Survey questions used to assess acute and broad bushfire exposure (yes/no), smoke exposure (not, minimally, mildly, moderately, severely), and acute and broad COVID-19 exposure (yes/no).

		Survey Questions
**Broad Bushfire** **Exposure**	**Acute Bushfire** **Exposure**	“Were you evacuated from your home or workplace because of the threat of fire?”
		“Were you personally involved in fighting bushfires threatening your own home or neighbourhood?”
		“Were your own home, possessions or workplace damaged or destroyed?”
		“Was your partner/spouse personally involved in fighting bushfires?”
		“Did you suffer any injury due to the fires?”
		“Did any relative or friend die or suffer injury due to the fires?”
		“Did you own any animal that suffered as a result of the fires?”
		“Did you feel very frightened or upset during the period of the fires?”
		“Apart from defending your own home and neighbourhood, did you do any work involving the bushfires and/or their effects? (e.g., fighting fires, keeping order, dealing with health effects, restoring power, caring for victims)”.
		“Were buildings in your suburb or area damaged or destroyed by fire?”,
		“Did any relative or friend have their home, possessions or workplace damaged or destroyed?”
		“Did you suffer significant financial loss due to the fires?”
		“Did the bushfires impact on your health care appointments?”
		“Did the bushfires impact on your access to medications/other health care supplies/baby care supplies?”
**Smoke** **Exposure**		During the period of 1st December 2019 to 14th of February 2020 what do you consider your bushfire smoke exposure was?”
**Broad COVID-19** **Exposure**	**Acute COVID-19 Exposure**	“I have been diagnosed positive for COVID-19”
		“I have a family member who has been diagnosed positive by a laboratory test”
		“I was directed by the health department to self-isolate”
		“I have a family member who is currently or has been required to self-isolate”
		“I have been tested for COVID-19—awaiting result”
		“I have been tested for COVID-19—negative result”
		“I have voluntarily self-isolated (current or previous)”
		“I have been a contact of someone who has been diagnosed positive by a laboratory test”
		“I know someone who has been or is currently required to self-isolate”

**Table 2 ijerph-21-00007-t002:** Participant characteristics at the start of the study stratified by pregnancy status at the time of the bushfires.

	Preconception (*n* = 66)	Pregnant (*n* = 636)	New Baby (*n* = 217)	Total (*n* = 919)	*p* Value
Age (years)					0.009
Mean (SD)	31.74 (4.63)	31.80 (4.42)	32.84 (4.30)	32.04 (4.43)	
Tertiary education	50 (75.8%)	472 (74.2%)	164 (75.6%)	686 (74.6%)	0.903
Middle/high income	62 (93.9%)	611 (96.1%)	213 (98.2%)	886 (96.4%)	0.193
Parity					0.916
Median (IQR)	1 (1)	1 (1)	1 (1)	1 (1)	
Prior depression	22 (33.3%)	108 (17.0%)	30 (13.8%)	160 (17.4%)	0.001
Prior anxiety	28 (42.4%)	220 (34.6%)	66 (30.4%)	314 (34.2%)	0.182
Wellbeing (WHO-5)					0.667
Mean (SD)	48.55 (19.51)	50.29 (20.26)	49.16 (20.77)	49.90 (20.31)	
DASS depression score					0.295
Mean (SD)	2.35 (3.08)	2.89 (3.36)	3.09 (3.59)	2.90 (3.40)	
DASS anxiety score					0.628
Mean (SD)	2.59 (2.95)	2.24 (2.71)	2.32 (3.12)	2.29 (2.83)	
DASS stress score					0.488
Mean (SD)	5.45 (3.90)	5.61 (4.16)	5.97 (4.32)	5.69 (4.18)	
DASS total score					0.621
Mean (SD)	10.39 (8.97)	10.75 (9.16)	11.39 (9.77)	10.87 (9.29)	
Acute bushfire exposure					0.782
Mean (SD)	1.06 (0.93)	1.15 (1.11)	1.12 (0.95)	1.13 (1.06)	
Broad bushfire exposure					0.655
Mean (SD)	1.97 (1.99)	2.21 (2.21)	2.24 (1.96)	2.20 (2.14)	
Smoke exposure					0.667
Not/mildly exposed	5 (7.6%)	44 (6.9%)	13 (6.0%)	62 (6.7%)	
Moderately exposed	18 (27.3%)	199 (31.3%)	58 (26.7%)	275 (29.9%)	
Severely exposed	43 (65.2%)	393 (61.8%)	146 (67.3%)	582 (63.3%)	
Acute COVID exposure					0.214
Mean (SD)	0.33 (0.57)	0.22 (0.47)	0.22 (0.45)	0.23 (0.47)	
Broad COVID exposure					0.139
Mean (SD)	1.38 (1.30)	1.11 (1.11)	1.21 (1.12)	1.15 (1.13)	

**Table 3 ijerph-21-00007-t003:** Comparison of DASS-21 total and sub-scales (depression, anxiety, stress) scores in the MC2020 study with scores in a population of pregnant mothers assessed in 2013 (Bryson et al., 2021 [50]), and Australian norms for the adult population (Crawford et al., 2011 [51]).

Study	Population	Depression	Dep SD	Anxiety	Anx SD	Stress	Str SD	Total Score	SD
MC2020	Preconception/pregnant/ post-partum women	2.9	3.4	2.3	2.8	5.7	4.2	10.9	9.3
Bryson et al.	Pregnant women	2.9	3.3	3.4	3.3	5.4	4.0	11.7	9.4
Crawford et al.	Adult Australian population	2.6	3.9	1.7	2.8	4.0	4.2	8.3	9.8

**Table 4 ijerph-21-00007-t004:** Associations between bushfire exposures (acute and broad) and mental health outcomes (WHO-5; DASS-21 depression, anxiety, and stress).

	WHO-5		Depression		Anxiety		Stress	
Intercept	54.722 ***	54.691 ***	5.131 ***	5.105 ***	6.588 ***	6.487 ***	7.350 ***	7.276 ***
	*p* < 0.001	*p* < 0.001	*p* < 0.00001	*p* < 0.00001	*p* < 0.001	*p* < 0.001	*p* < 0.001	*p* < 0.001
Age, yrs	−0.033	−0.039	−0.094 ***	−0.092 ***	−0.140 ***	−0.137 ***	−0.103 **	−0.100 **
	*p* = 0.841	*p* = 0.816	*p* = 0.001	*p* = 0.001	*p* < 0.001	*p* < 0.001	*p* = 0.003	*p* = 0.003
Parity, number	−1.599	−1.541	0.439 **	0.423 **	0.535 ***	0.516 ***	0.615 ***	0.588 ***
	*p* = 0.062	*p* = 0.073	*p* = 0.002	*p* = 0.003	*p* = 0.00001	*p* = 0.00001	*p* = 0.0004	*p* = 0.001
Tertiary education (yes)	1.605	1.507	−0.684 **	−0.656 *	−0.901 ***	−0.869 ***	−0.824 *	−0.777 *
	*p* = 0.331	*p* = 0.363	*p* = 0.012	*p* = 0.016	*p* = 0.00004	*p* = 0.0001	*p* = 0.013	*p* = 0.020
Pregnancy phase (pregnant)	1.889	1.952	0.515	0.499	−0.368	−0.383	0.105	0.079
	*p* = 0.470	*p* = 0.456	*p* = 0.229	*p* = 0.243	*p* = 0.287	*p* = 0.266	*p* = 0.842	*p* = 0.880
Pregnancy phase (new baby)	0.774	0.919	0.837	0.803	−0.124	−0.152	0.602	0.549
	*p* = 0.786	*p* = 0.747	*p* = 0.072	*p* = 0.085	*p* = 0.741	*p* = 0.686	*p* = 0.291	*p* = 0.335
Bushfire exposure (acute)	−1.826 **		0.390 ***		0.214 *		0.565 ***	
	*p* = 0.006		*p* = 0.0003		*p* = 0.013		*p* = 0.00002	
*Bushfire exposure (broad)*		*−0.902 ***		*0.203 ****		*0.138 ***		*0.304 ****
		*p* = 0.006		*p* = 0.0002		*p* = 0.002		*p* = 0.00001
Smoke exposure (moderate)	−2.552	−2.528	−0.171	−0.181	0.479	0.459	0.888	0.867
	*p* = 0.371	*p* = 0.375	*p* = 0.714	*p* = 0.697	*p* = 0.204	*p* = 0.222	*p* = 0.120	*p* = 0.128
Smoke exposure (severe)	−3.898	−3.793	0.035	0.002	0.592	0.545	1.078 *	1.018
	*p* = 0.152	*p* = 0.165	*p* = 0.938	*p* = 0.998	*p* = 0.100	*p* = 0.130	*p* = 0.048	*p* = 0.062
Observations	919	919	919	919	919	919	919	919
Log Likelihood	−4061.52	−4061.64	−2396.80	−2396.29	−2201.50	−2199.47	−2583.26	−2581.78
Akaike Inf. Crit.	8141.04	8141.28	4811.60	4810.58	4421.00	4416.93	5184.52	5181.57

Note: Bonferroni corrected *p*-value thresholds set at *p* <0.0125. * *p* < 0.05; ** *p* < 0.0125; *** *p* < 0.001.

## Data Availability

The data presented in this study are available on request from the corresponding author. The data are not publicly available due to ethical and consent restrictions.

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
