# Peer review of "Bushfires and Mothers’ Mental Health in Pregnancy and Recent Post-Partum"

_ijerph, 2023, doi:10.3390/ijerph21010007_

Round 1
Reviewer 1 Report
Comments and Suggestions for Authors
This is an interesting and well written report on a survey study of bushfire exposure and COVID-19 pandemic exposure in relation to mental health related to three time periods of pregnancy. A few points of clarification are needed:
1.Please provide some additional information about the “adult population norms” used as a comparison (i.e. Table 2). What is the age range/distribution for these values and do they pertain to females only or to a male/female population? P. 10, line 340 references “Australian norms for this age group” but it’s not clear exactly how well these norms overlap the study population.
To the extent that they may differ in sex or age distribution from they study population, this may impact comparisons and should be more clearly noted.
2.Table 2 title “Comparison of DASS-21 total and sub-scales (depression, anxiety, stress) scores 235 in the MC2020 study with scores in a comparable population of pregnant mothers ….” As pointed out elsewhere by the authors, the population of Bryson et al had important differences from the study population and thus it would be more appropriate if the word “comparable” was removed from the description.
3.Some context would be helpful in terms of representativeness of the sample. Participants were recruited by “mainstream media, social media and health service providers” and it is likely to be a highly self selected sample. For example, could the demographics of the pregnant/post partum sample be compared with vital statistics of the same geographic area/time period?
4.It is a limitation that power may not have been sufficient to detect interactions.
Author Response
R1
This is an interesting and well written report on a survey study of bushfire exposure and COVID-19 pandemic exposure in relation to mental health related to three time periods of pregnancy. A few points of clarification are needed:
Response
Thanks you for your review and positive appraisal, we have carefully addressed the points you have raised below.
Point 1
Please provide some additional information about the “adult population norms” used as a comparison (i.e. Table 2). What is the age range/distribution for these values and do they pertain to females only or to a male/female population? P. 10, line 340 references “Australian norms for this age group” but it’s not clear exactly how well these norms overlap the study population.
To the extent that they may differ in sex or age distribution from they study population, this may impact comparisons and should be more clearly noted.
Response:
This information has been added in text (page 7 lines 254-258)
Point 2
Table 2 title “Comparison of DASS-21 total and sub-scales (depression, anxiety, stress) scores 235 in the MC2020 study with scores in a comparable population of pregnant mothers ….” As pointed out elsewhere by the authors, the population of Bryson et al had important differences from the study population and thus it would be more appropriate if the word “comparable” was removed from the description.
Response:
The legend of Table 2 has been amended as suggested.
Point 3
Some context would be helpful in terms of representativeness of the sample. Participants were recruited by “mainstream media, social media and health service providers” and it is likely to be a highly self selected sample. For example, could the demographics of the pregnant/post partum sample be compared with vital statistics of the same geographic area/time period?
Response:
As the reviewer notes, although the research team made a great effort to develop and implement a recruitment strategy as inclusive as possible, it is still likely that it involved some selection bias and that our sample is not fully representative of the targeted population.
To provide context we have used data from the Australian Bureau of Statistics (ABS) 2022 “Education and Work” and 2021 “Employee Earnings and Hours” surveys including women between 25-34 years living in the Australian Capital Territory. Compared to ABS estimates included participants had a lower proportion of tertiary education (64.4% vs 69.5%), a greater proportion of middle/high income (96.4% vs 73.4%), and higher parity (0.73 vs 0.57). These figures suggest a moderate amount of bias which in our view makes our sample somewhat but not fully comparable to the population from which it was drawn. We have included these points in the revised manuscript (page 5, lines 235-246) and in the limitation section (page 13, 451-453).
Point 4
It is a limitation that power may not have been sufficient to detect interactions.
Response:
The statistical power for interactions in regression analyses is notoriously challenging to estimate, but we agree with the point made by this reviewer and we have added it to the limitation section (page 13, 464-465).
Reviewer 2 Report
Comments and Suggestions for Authors
The manuscript titled “Bushfires and mothers’ mental health in pregnancy and recent post-partum” aimed to investigate the impact of the 2019/2020 Australian bushfires exposure on the peripartum women’s mental health. This study may be helpful to understand how bushfires impact the mental health of peripartum women. However, there are several concerns in the manuscript.
Comments
#1. In the introduction section, the authors made little mention of the associations between natural disasters and mental health. There is plenty of literature about this topic. What is already known about this topic?
#2. Online surveys inevitably encounter the problem of fraudulent responses. Did the authors ask questions aimed at identifying them?
#3. Did the authors pay any compensation to the subjects? Explain it.
#4. In the results section, the authors compared the scores of mental health of the subjects to the Australian adult population norms. However, the Australian adult population norms examined before COVID-19 (in 2011). Due to the COVID-19 pandemic, the psychiatric conditions among them might be different from then, regardless of the bushfires. The authors should provide a more detailed discussion on the psychological impacts of the COVID-19 pandemic. The following articles may be useful:
Reference:
Takubo Y et al. Psychological impacts of the COVID-19 pandemic on one-month postpartum mothers in a metropolitan area of Japan. BMC Pregnancy Childbirth. 2021 Dec 28; 21(1): 845.
Takubo Y et al. Changes in thoughts of self-harm among postpartum mothers during the prolonged COVID-19 pandemic in Japan. Psychiatry Clin Neurosci. 2022 Jul 13.
#5. There are generally few references to support the discussion. Cite references appropriately to reinforce the discussion.
#6. The authors described in the discussion section as follows, “However, a similar effect was not observed for those who experienced smoke exposure. This is somewhat surprising because it might have been expected that pregnant women would be particularly concerned about the effect smoke inhalation could have on the health and development of their baby.” The authors may discuss the reason why.
#7. Please check the descriptive format of all references. The following references seem to contain typos.
1. 2021, I.P.o.C.C. The Physical Science Basis. Contribution of Working Group I to the Sixth Assessment Report. . 2021.
14. da Silva, A.M.C.; Moi, G.P.; Mattos, I.E.; Hacon, S.D. Low birth weight at term and the presence of fine particulate matter and carbon monoxide in the Brazilian Amazon: a population-based retrospective cohort study. Bmc Pregnancy Childb 2014, 14, doi:Artn 309 10.1186/1471-2393-14-309.
Author Response
R2
The manuscript titled “Bushfires and mothers’ mental health in pregnancy and recent post-partum” aimed to investigate the impact of the 2019/2020 Australian bushfires exposure on the peripartum women’s mental health. This study may be helpful to understand how bushfires impact the mental health of peripartum women. However, there are several concerns in the manuscript.
Point 1.
In the introduction section, the authors made little mention of the associations between natural disasters and mental health. There is plenty of literature about this topic. What is already known about this topic?
Response:
As requested we have now included more background on the associations between natural disasters and mental health in the introduction (page 2, lines 62-70).
Point 2
Online surveys inevitably encounter the problem of fraudulent responses. Did the authors ask questions aimed at identifying them?
Response:
We have been involved in the design, management, and administration of large population surveys for more than 15 years. In our experience, fraudulent responses typically occur when a financial incentives are provided but are not a substantive issue when participants are invited to volunteer to progress knowledge of important topics relevant to them through community and health networks. As a financial incentive was not offered a more relevant risk may be partial completion. Since completion rates were high, participants were highly motivated due to their concern for the health of their babies, and since we only included participants for whom the key data were available we are confident that the data analysed is of high quality. Nonetheless, all variables were checked carefully and we did not identify evidence of serial or systematically aberrant responses.
In addition, while we did not include specific questions aimed at detecting careless responding such as “If you read this question select response X”, the Positive and Negative Impression Management scales (PIM/NIM) which are part of the Personality Assessment Inventory (PAI) and can assist in detecting socially desirable and exaggerated responding were included in one of the surveys. To address this query we used them to identify participants who scored highly on these scales (PIM>9: n=42; NIM>5: n=7) and conducted sensitivity analyses excluding them as well as exluding participants for whom data on these scales were not available (n=203). The results remained essentially the same with some marginal differences in effect sizes but not significance. We are therefore confident of the quality and robustness of the participants’ responses.
Point 3
Did the authors pay any compensation to the subjects? Explain it.
Response:
No compensation was offered. We have now stipulated this point in the revised manuscript (page 3, line 123-124).
Point 4
In the results section, the authors compared the scores of mental health of the subjects to the Australian adult population norms. However, the Australian adult population norms examined before COVID-19 (in 2011). Due to the COVID-19 pandemic, the psychiatric conditions among them might be different from then, regardless of the bushfires. The authors should provide a more detailed discussion on the psychological impacts of the COVID-19 pandemic. The following articles may be useful:
Reference:
Takubo Y et al. Psychological impacts of the COVID-19 pandemic on one-month postpartum mothers in a metropolitan area of Japan. BMC Pregnancy Childbirth. 2021 Dec 28; 21(1): 845.
Takubo Y et al. Changes in thoughts of self-harm among postpartum mothers during the prolonged COVID-19 pandemic in Japan. Psychiatry Clin Neurosci. 2022 Jul 13.
Response:
Thank you for these suggestions and the references. We have discussed this confounding issue in the revised manuscript and included the first of these suggested references (page 12 lines 409-415).
Point 5
There are generally few references to support the discussion. Cite references appropriately to reinforce the discussion.
Response:
As suggested, we have included more supporting references throughout the discussions.
Point 6
The authors described in the discussion section as follows, “However, a similar effect was not observed for those who experienced smoke exposure. This is somewhat surprising because it might have been expected that pregnant women would be particularly concerned about the effect smoke inhalation could have on the health and development of their baby.” The authors may discuss the reason why.
Response:
As suggested, we have added a discussion of the possible origin of these surprising results (page 12, lines 381-392).
Point 7
Please check the descriptive format of all references. The following references seem to contain typos.
- 2021, I.P.o.C.C. The Physical Science Basis. Contribution of Working Group I to the Sixth Assessment Report. . 2021.
- da Silva, A.M.C.; Moi, G.P.; Mattos, I.E.; Hacon, S.D. Low birth weight at term and the presence of fine particulate matter and carbon monoxide in the Brazilian Amazon: a population-based retrospective cohort study. Bmc Pregnancy Childb 2014, 14, doi:Artn 309 10.1186/1471-2393-14-309.
Response:
We have checked all references and corrected them accordingly.
Reviewer 3 Report
Comments and Suggestions for Authors
The authors make an original contribution by trying to determine the impact of direct and indirect exposure to bushfires on the mental health of pregnant women and women with newborns.
The topic of mental health in pregnant women is very interesting and I am convinced that they can make relevant contributions to this topic with their data. However, I believe that your manuscript is not suitable for publication in the way it is written and that you should make substantial changes in content. I expected to find more information when I read bushfires in the title but then in methodology I see that this is added as a supplementary file and in the results not all the information is there. Likewise, I do not think it is rigorous to analytically compare your data with those of other studies on results.
Author Response
R3
The authors make an original contribution by trying to determine the impact of direct and indirect exposure to bushfires on the mental health of pregnant women and women with newborns.
The topic of mental health in pregnant women is very interesting and I am convinced that they can make relevant contributions to this topic with their data. However, I believe that your manuscript is not suitable for publication in the way it is written and that you should make substantial changes in content. I expected to find more information when I read bushfires in the title but then in methodology I see that this is added as a supplementary file and in the results not all the information is there. Likewise, I do not think it is rigorous to analytically compare your data with those of other studies on results.
Response:
Thank you for noting the original contribution we are making with this research and highlighting the importance of the health of pregnant women. We are sorry that you found that providing part of the content in supplementary files made our paper less accessible. It should be noted however, that for every reviewer who asks for more content in the manuscript there are others who request that more content be moved to the supplementary material to improve the flow of the paper. In addition, many journals express strong preference that the details of the methods and results be moved to the supplementary material to decrease manuscript length. Nonetheless, as requested we have moved more content back to the main manuscript which we hope will satisfy this reviewer.
We also acknowledge that in an ideal world it would have been better to compare our findings to norms developed in an identical population (i.e. pregnant women and new mothers living in the same region with the same characteristics). However, given the lack of such norms we have tried hard to contextualise our findings in light of more general norms and the results of large observational studies which provide some metrics to inform interpretation. Admittedly the comparison is not perfect and we have acknowledged this in the discussion and limitations and outlined some key differences between the populations compared. This is not an unusual approach and in fact it is widely applied in systematic reviews and meta-analyses.
Round 2
Reviewer 2 Report
Comments and Suggestions for Authors
Thank you for addressing all the comments. The manuscript has been improved and my recommendation is to accept it in its current form.